# Potential Role of the mTORC1-PGC1α-PPARα Axis under Type-II Diabetes and Hypertension in the Human Heart

**DOI:** 10.3390/ijms24108629

**Published:** 2023-05-11

**Authors:** Tianyu Hang, Jairo Lumpuy-Castillo, Naroa Goikoetxea-Usandizaga, Mikel Azkargorta, Gonzalo Aldámiz, Juan Martínez-Milla, Alberto Forteza, José M. Cortina, Jesús Egido, Félix Elortza, Malu Martínez-Chantar, José Tuñón, Óscar Lorenzo

**Affiliations:** 1Laboratory of Diabetes and Vascular Pathology, IIS-Fundación Jiménez Díaz, Universidad Autónoma, 28040 Madrid, Spain; 2Biomedical Research Network on Diabetes and Associated Metabolic Disorders (CIBERDEM), Carlos III National Health Institute, 28029 Madrid, Spain; 3Liver Disease Lab, Center for Cooperative Research in Biosciences (CIC bioGUNE), Basque Research and Technology Alliance (BRTA), 48160 Derio, Spain; 4Biomedical Research Network on Liver and Digestive Diseases (CIBERehd), Carlos III National Health Institute, 28029 Madrid, Spain; 5Proteomics Platform, Center for Cooperative Research in Biosciences (CIC bioGUNE), Basque Research and Technology Alliance (BRTA), 48160 Derio, Spain; 6Cardiovascular Surgery Department, Fundación Jiménez Díaz Hospital, 28040 Madrid, Spain; 7Cardiology Department, Fundación Jiménez Díaz Hospital, 28040 Madrid, Spain; 8Cardiovascular Surgery Department, Doce de Octubre Hospital, 28041 Madrid, Spain; 9Medicine Department, Universidad Autónoma, 28029 Madrid, Spain; 10Biomedical Research Network on Cardiovascular Diseases (CIBERCV), Carlos III National Health Institute, 28029 Madrid, Spain

**Keywords:** type-II diabetes, hypertension, cardiomyopathy, mTOR complexes

## Abstract

Type-2 diabetes (T2DM) and arterial hypertension (HTN) are major risk factors for heart failure. Importantly, these pathologies could induce synergetic alterations in the heart, and the discovery of key common molecular signaling may suggest new targets for therapy. Intraoperative cardiac biopsies were obtained from patients with coronary heart disease and preserved systolic function, with or without HTN and/or T2DM, who underwent coronary artery bypass grafting (CABG). Control (*n* = 5), HTN (*n* = 7), and HTN + T2DM (*n* = 7) samples were analysed by proteomics and bioinformatics. Additionally, cultured rat cardiomyocytes were used for the analysis (protein level and activation, mRNA expression, and bioenergetic performance) of key molecular mediators under stimulation of main components of HTN and T2DM (high glucose and/or fatty acids and angiotensin-II). As results, in cardiac biopsies, we found significant alterations of 677 proteins and after filtering for non-cardiac factors, 529 and 41 were changed in HTN-T2DM and in HTN subjects, respectively, against the control. Interestingly, 81% of proteins in HTN-T2DM were distinct from HTN, while 95% from HTN were common with HTN-T2DM. In addition, 78 factors were differentially expressed in HTN-T2DM against HTN, predominantly downregulated proteins of mitochondrial respiration and lipid oxidation. Bioinformatic analyses suggested the implication of mTOR signaling and reduction of AMPK and PPARα activation, and regulation of PGC1α, fatty acid oxidation, and oxidative phosphorylation. In cultured cardiomyocytes, an excess of the palmitate activated mTORC1 complex and subsequent attenuation of PGC1α-PPARα transcription of β-oxidation and mitochondrial electron chain factors affect mitochondrial/glycolytic ATP synthesis. Silencing of PGC1α further reduced total ATP and both mitochondrial and glycolytic ATP. Thus, the coexistence of HTN and T2DM induced higher alterations in cardiac proteins than HTN. HTN-T2DM subjects exhibited a marked downregulation of mitochondrial respiration and lipid metabolism and the mTORC1-PGC1α-PPARα axis might account as a target for therapeutical strategies.

## 1. Introduction

The increasing incidence of heart failure is mostly attributable to recognized ischemic heart disease and to the direct effects of its major risk factors, such as type 2 diabetes (T2DM) and arterial hypertension (HTN), on myocardial structure and function [1]. Patients with T2DM often exhibit multiple comorbidities. The highest co-prevalence was demonstrated for the combination of HTN and hyperlipidemia (67.5%), followed by overweight/obesity and HTN (66.0%) and overweight/obesity and hyperlipidemia (62.5%) [2]. Interestingly, T2DM, obesity, and HTN may act synergistically to produce coronary heart disease, left ventricle (LV) dysfunction, and heart failure [3,4]. However, the underlying cellular and molecular mechanisms are not fully elucidated, which may contribute to the absence of specific and valid treatments for these subjects.

The failure of insulin sensitivity and glucose assimilation in T2DM hearts promotes fatty acids (i.e., palmitate) as unique energetic substrates for ATP generation [5]. Non-assimilated glucose led to glucose-derived metabolites and activates pro-oxidant and inflammatory pathways [6], and excessive utilization of fatty acid saturates β-oxidation and produces lipid metabolites with pro-oxidant and pro-apoptotic properties [7]. In addition, the oxidation of fatty acids consumes more oxygen per ATP produced than glucose, and this could be increased in hyperlipidemic and obese patients [8,9]. Importantly, T2DM-associated glucotoxicity and lipotoxicity trigger early alterations in mitochondrial performance [10,11]. In this line, HTN is also a major promoter of left ventricular remodeling through pro-hypertrophic/fibrotic actions and severe alterations in metabolic substrate utilization and mitochondrial function [12]. Therefore, cardiac mitochondrial dysfunction may represent a common cellular alteration in T2DM and HTN that could be synergistically enforced when they coexist [2,13].

The discovery of key molecular pathways involved in mitochondrial regulation under T2DM and HTN may suggest new targets for therapeutic strategies. Herein, we reveal potential mediators of mitochondrial failure by proteomics and bioinformatics analysis in cardiac biopsies of T2DM patients with/without HTN and preserved systolic function. The lipid component and the implication of the mTORC1-PGC1α-PPARα axis was further examined in cultured cardiomyocytes. This molecular signaling could provide interesting targets for early attenuation of coronary heart disease in these patients.

## 2. Results

### 2.1. Characterization of the Cardiomyopathy Population

This is a study investigating differences between HTN and HTN + T2DM in coronary heart disease with preserved ejection fraction. The HTN group included seven patients undergoing coronary artery bypass grafting (CABG) with an average age of 64.7 ± 7.7 years (Table 1A,B). Six of them were male and the body mass index (BMI) was 29.4 ± 6.6 kg/m^2^. The ejection fraction was 56.9 ± 5.1%, and the diastolic diameter was 42.1 ± 4.5 mm. One patient had LV hypertrophy, and plasma creatinine was 0.96 ± 0.24 mg/dL. The group with both HTN and T2DM included seven patients undergoing CABG. The average age was 72.5 ± 8.8 years, six of them were male, and the BMI was 32.8 ± 5.3 kg/m^2^ (Table 1A,B). The ejection fraction was 59.1 ± 5.7% and the diastolic diameter was 44.0 ± 3.2 mm. Two patients had LV hypertrophy, and plasma creatinine was 1.07 ± 0.4 mg/dL. The control group included one patient undergoing CABG and four patients undergoing valve replacement (two mitral and aortic, one mitral, and one aortic). The age was 71.2 ± 8.4 years, two patients were male, and the BMI was 29.5 ± 4.8 kg/m^2^ (Table 1A,B). The ejection fraction was 57.0 ± 6.7% and the diastolic diameter was 45.6 ± 9.9 mm. Two patients had LV hypertrophy and the average plasma creatinine was 0.76 ± 0.15 mg/dL. Altogether, the three groups were well balanced, without significant differences in age (*p* = 0.21), sex distribution (*p* = 0.27), BMI (*p* = 0.72), ejection fraction (*p* = 0.73), diastolic diameter (*p* = 0.60), presence of LV hypertrophy (*p* = 0.64), and creatinine plasma levels (*p* = 0.26). However, the lipid profile was altered in the HTN and HTN-T2DM groups (Table 1B). In the former, the low-density lipoproteins (LDL-c) were elevated (83.1 mg/dL), whereas the high-density lipoprotein (HDL-c) levels were low (<40 mg/dL). In the latter, patients showed marked dyslipidemia with increased LDL-c, triglycerides (TG), and total cholesterol (TC), but reduced HDL-c.

In addition, ACEI/ARB were used in 85.7%, 57.1%, and 80%; β-blockers in 100%, 57.1%, and 80%; MRA in 28.6%, 0%, and 40%; and statins in 100%, 100%, and 40% of patients in the HTN, HTN-T2DM, and control groups, respectively (Table 1C). Regarding the antithrombotic therapy, in the HTN group, 57.1% of patients received aspirin, 28.6% of them received Dual antiplatelet therapy (DAPT), and 14.3% received acenocumarol. In the HTN-T2DM group, 55.6% received aspirin, 33.3% received DAPT, and 11.1% did not receive antithrombotic therapy. In the control group, 20% of subjects received DAPT and 80% received acenocumarol. Nevertheless, HTN could not be controlled in 29.6% of subjects in the HTN group and in 78.8% of patients in the HTN-T2DM group, though blood glucose concentrations were maintained below non-pathological levels in HTN-T2DM individuals (Table 1B).

### 2.2. Cardiac Proteomics Patterns after HTN and HTN-T2DM

Intraoperative biopsies from the interventricular septum were isolated from the three groups of patients for label-free nano LC-MS/MS proteomic analysis. A total of 1287 proteins were identified in human hearts, of which 677 proteins were significantly altered among groups (*p* < 0.05). Additionally, 666 proteins were differentially expressed in HTN-T2DM patients compared to control (117 were upregulated and 549 were downregulated) (Appendix A), while only 45 proteins were found changed in HTN compared to control (4 were overexpressed and 41 were diminished) (Appendix A). After filtering the proteins with non-cardiac expression (i.e., blood and liver proteins), we found 529 differential proteins in HTN-T2DM subjects compared to control (Figure 1A), among which, 49 were upregulated and 480 were downregulated. Most of them, 428, were distinct from the HTN group. In this regard, HTN patients showed 41 cardiac proteins altered against control; 3 of them were overexpressed, while 38 were downregulated. Most of these factors, 39, were overlapped with the HTN-T2DM group; 3 upregulated proteins (i.e., annexin A2, myosin-6, and tripeptidyl-peptidase-2) and 36 downregulated. Two additional proteins were specifically reduced in HTN patients (i.e., 26S-protease regulatory subunit-6 and asparagine-tRNA ligase). More interestingly, 78 changed proteins were observed to be differentially expressed in HTN-T2DM versus HTN (Figure 1A), among which 15 were upregulated and 63 were reduced; six of them specifically changed in HTN-T2DM (Appendix A).

Thus, according to proteomics, HTN-T2DM may induce more severe effects on cardiac protein expression than HTN alone. Most HTN-T2DM-associated alterations (81% of proteins) were not observed in HTN, suggesting additional changes when both pathologies coexist. Of note, these modifications were induced even under preserved systolic function.

### 2.3. Alteration of Factors in HTN-T2DM and HTN Hearts

After clustering the differentially expressed proteins in functional states by using the Uniprot database (Appendix A), we unveiled in HTN-T2DM hearts a predominance of deregulated proteins related to metabolism (229 proteins: 43.3%) (Figure 1B, top), particularly in mitochondrial respiration (96; 42.0%), lipid (40; 17.5%) and carbohydrate (32; 14.0%) metabolism, pyruvate decarboxylation/tricarboxylic acid cycle (TCA) (30; 13.1%), and amino-acid metabolism (31; 13.4%). HTN-T2DM hearts also exhibited mostly increased proteins in cytoskeleton regulation and fibrosis (138; 26.1%), RNA/Protein synthesis (46; 8.7%), and inflammation/apoptosis (39; 7.4%), and reduced factors for mitochondrial homeostasis (33; 6.3%) (Figure 1B, top). In contrast, HTN showed an augmented protein expression in RNA/Protein synthesis (19; 43.3%) and a decreased expression in metabolic proteins (8; 19.5%), which included factors related to mitochondrial respiration (2; 25%), carbohydrate (2; 25%) and amino-acid metabolism (3; 37.5%), and pyruvate decarboxylation/TCA (1; 12.5%) (Figure 1B, middle). Additionally, these hearts increased factors related to inflammation/apoptosis (5; 12.3%) and cytoskeleton regulation and fibrosis (3; 7.3%), and diminished proteins linked with mitochondrial homeostasis (3; 7.3%).

Interestingly, by comparing HTN-T2DM versus HTN (Figure 1B, bottom), HTN-T2DM exhibited a significant downregulation of metabolic proteins (38; 48.7%), including those for mitochondrial respiration (21; 55.2%), lipid metabolism (10; 26.3%), pyruvate decarboxylation/TCA (4; 10.5%), and amino-acid metabolism (3; 7.9%). Again, there was an upregulation in cytoskeleton regulation and fibrosis (16; 20.5%), inflammation/apoptosis (9; 11.5%), and RNA/protein synthesis (9; 11.5%), and a reduction of mitochondrial homeostasis factors (4; 5.1%) (Figure 1B, bottom). Therefore, the coexistence of T2DM and HTN might induce a greater reduction in cardiac proteins mainly involved in metabolism, particularly in mitochondrial respiration and lipid utilization.

### 2.4. Activation of Molecular Pathways in HTN-T2DM and HTN Hearts

To integrate and interpret the data obtained from proteomics, we used the bioinformatics tool IPA^®^. The altered proteins and their expression levels were analyzed in HTN-T2DM and HTN hearts to find candidate molecular pathways and upstream regulators. Following the manufacturer’s instructions [17], we considered only z-score variations higher than 2.0-fold. In consonance with proteomics, HTN-T2DM could promote the inhibition of glycolysis and gluconeogenesis (z-score = −3.3) and the PDH complex (z-score = −2.2), which may affect the TCA cycle (z-score = −4.1) (Table 2). Additionally, it might induce a reduction in AMPK (z-score < −2.8) and fatty acid α- and β-oxidation (z-score < −2), ketogenesis (z-score = −2), and oxidative phosphorylation (z-score < −8) (Table 2). The eukaryotic initiation factor-2 signaling (eIF2) as an mTOR interactor and regulator of mRNA translation at the ribosomal level would have also been inhibited (z-score = −3.1). In consequence, both the acute phase response signaling and the sirtuin signaling might be activated (z-score > 2.6). However, for HTN hearts, bioinformatics only predicted the inhibition of eIF2 (z-score = −2.4) as a candidate involved pathway (Table 2).

Furthermore, the IPA^®^ suggested inhibition of upstream regulators such as PPARγ and PPARα transcription factors (z-score < −3.1) and anti-inflammatory proteins CD28, CD3, and NFE2L2 (z-score < −3.9) (Table 3) for HTN-T2DM hearts. Additionally, it indicated a further decrease in MYCN and MYC transcription factors (z-score < −5.5) associated with proliferation, glycolysis, and mitochondrial biogenesis, and of PPARγ coactivator-1 (PGC1α and the insulin receptor (z-score < −6). In contrast, the pro-apoptotic CD437 (z-score = 5.4) could be activated. Consequently, the HTN-T2DM hearts might upregulate the anti-oxidative mediator MAP4K4 (z-score = 6.1) and the cardioprotective subunit of mTORC2, Rictor (z-score = 8.7). In the case of HTN hearts, bioinformatics also proposed an inhibition of MYCN and MYC (z-score < −2.7) and of anti-oxidative/-inflammatory factors (CD28 and CD3, z-score = −2), while suggesting an enhancement of Rictor and CD437 (z-score > 2) and the anti-oxidative mediator NUPR1 (z-score = 2) (Table 3).

Altogether, HTN-T2DM hearts might lead to reduced glycolysis and gluconeogenesis and decreased AMPK and fatty acid oxidation, which would induce significant damage to mitochondria (i.e., oxidative phosphorylation, PGC1α, MYC), increased inflammation (CD28, NFEL2), and adaptive cardioprotective responses including Rictor activation. For HTN hearts, bioinformatics suggested a stimulation of oxidative/inflammatory responses and Rictor. Therefore, mTOR complexes and AMPK may be crucial for these HTN-T2DM and HTN hearts under coronary heart disease and preserved systolic function.

### 2.5. The Excess of Lipids as a Major Activator of mTORC1 in Cardiomyocytes

Next, we tested in vitro whether major mediators of HTN and T2DM such as angiotensin-II (AngII) and high concentrations of glucose (HG) and/or saturated fatty acids (HFA) might regulate the mTOR and AMPK signaling. In cultured cardiomyocytes, 24 h stimulation with HFA (90 μM), but not with AngII or HG, increased the Thr-421/Ser-424 phosphorylation of the main downstream mediator of the mTORC1 complex, p70-S6K (Figure 2A). Additionally, pre-treatment with an acute inhibitor of mTORC1 at nanomolar concentrations [18], rapamycin, abolished this effect. These actions were also observed for HFA at 150 μM (Appendix A), and the addition of AngII (or HG) to HFA did not further increase the p70-S6K phosphorylation (Figure 2B). Moreover, a specific activator of mTOR (the common subunit for mTORC1 and mTORC2), MHY1485, enhanced this HFA-dependent p70-S6K activation (Figure 3A).

In addition, previous reports have established a connection between mTORC1 activation and AMPK signaling [19,20]. In fact, in cardiomyocytes, we observed that the HFA-induced phosphorylation of p70-S6K was blocked after preincubation with an AMPK activator, metformin (Figure 3A and Appendix A). Additionally, specific AMPK activation by phosphorylation at Thr-172 was not increased after HFA (90–150 μM) nor after both rapamycin and HFA stimulation (Figure 3B). As expected, phospho-AMPK^Thr172^ was elevated after metformin but not after silencing the mTORC2 subunit, Rictor. Thus, HFA may stimulate mTORC1 independently of AMPK, though AMPK activation could regulate the mTORC1 signaling induced by HFA.

### 2.6. Activation of mTORC1 and Regulation of PGC1α-PPARα Signaling after Excessive Fatty Acid Stimulation

Interestingly, activation of mTORC1 by HFA could influence mitochondrial function [19,21], and PGC1α, a key factor for mitochondrial metabolism and biogenesis, could be sensitive to mTORC1/AMPK signaling [22,23,24]. In fact, bioinformatics predicted a decrease of PGC1α in HTN-T2DM hearts (Table 3). Interestingly, in HFA-stimulated cardiomyocytes, PGC1α was also downregulated, and this effect was reverted by rapamycin and recovered with MHY1485 (Figure 4A,B). Therefore, HFA could reduce PGC1α levels by the specific mediation of mTORC1, and not by mTORC2. In this regard, pre-treatment with siRictor did not modify PGC1α levels after HFA (Figure 4B).

In addition, PGC1α can coactivate and regulate the expression of specific transcription factors such as PPARα. Then, the PGC1α-PPARα complex may increase gene expression for lipid utilization and mitochondrial metabolism [25,26]. In this line, ACADm, a key β-oxidation enzyme, was reduced after HFA (Figure 5A), confirming data from proteomics (Appendix A). These levels were also partially recovered with rapamycin or metformin preincubation (Figure 5A), but not by MHY1485. Likely, the reduction of PGC1α induced by HFA-mediators mTORC1 and AMPK may affect ACADm levels in cardiomyocytes. In this sense, PPARα transcripts were also decreased after HFA and partially returned by rapamycin and metformin (Figure 5B). In addition, a key subunit of the catalytic core of the mitochondrial succinate dehydrogenase complex II (SDHB), which was also found lessened by proteomics (Appendix A), was diminished after HFA. Again, this action was attenuated by rapamycin and metformin, but not affected by Rictor or PGC1α silencing (Figure 5B).

### 2.7. Downregulation of PGC1α Signaling and ATP Formation after Excessive Fatty Acid Stimulation

Finally, decreased levels of PGC1α and specific factors related to β-oxidation and the mitochondrial ETC (i.e., PPARα, ACADm, SDHB) in the HTN-T2DM hearts could be reflected in ATP production. Cellular ATP can be synthesized through glycolysis in the cytosol or through oxidative phosphorylation at the mitochondria. By using a metabolic analyzer (i.e., XF Seahorse), HFA did not modify the total ATP formation but significantly decreased mitochondrial ATP synthesis (Figure 6A) and the baseline OCR (Figure 6B). This total ATP formation was also confirmed by a bioluminescence assay (Appendix A). Indeed, the ratio between mitochondrial and glycolytic ATP production was significantly diminished after HFA (Figure 6C). Interestingly, by silencing PGC1α in HFA-stimulated cells, the total and both mitochondrial (with baseline OCR) and glycolytic ATP generation were further reduced (Figure 6A,B). A similar effect was seen after preincubation with a PGC1α inhibitor, SR-18292. Additionally, pre-treatment with rapamycin and siRictor did not significantly alter HFA-induced ATP formation and the related OCR (Figure 6A,B), but siPGC1α, SR-18292, and siRictor alone decreased the mitochondrial/glycolytic ATP ratio (Appendix A). As expected, preincubation with metformin, a known inhibitor of mitochondrial complex I [27], further lessened mitochondrial ATP formation and baseline OCR while enhancing glycolytic ATP (Figure 6A–C). Therefore, stimulation of cardiomyocytes with HFA at non-apoptotic conditions (90 μM, 24 h) might unbalance mitochondrial/glycolytic ATP generation in a PGC1α-dependent manner. In this regard, we tested whether an activator of PGC1α might affect the downregulated expression of HFA-induced PPARα and SDH genes. Remarkably, we observed by qPCR that PPARα and SDHB were increased after ZLN005 in cardiomyocytes stimulated with HFA (Figure 6D). These data reinforce the potential key role of PGC1α on mitochondrial bioenergy in HTN-T2DM, particularly in an excessive fatty acid environment.

## 3. Discussion

Patients with coronary heart disease and HTN-T2DM produce greater alterations in cardiac protein content than those with coronary heart disease and HTN. Previous reports suggested that the existence of insulin resistance, impairment of endothelial function, oxidative stress, and mitochondrial dysfunction may be responsible for this synergistic impact [12,28,29,30]. Herein we describe that coincidence of HTN and T2DM induces more than ten times the number of differentially expressed factors and higher fold changes of protein expression than HTN alone, even under a preserved systolic function and after pharmacological therapies.

Under HTN, cardiac tissue may suffer from overactivation of the renin–angiotensin–aldosterone (RAA) and sympathetic nervous systems, oxidation/inflammation, and altered T-cell function, among other mechanisms [31]. As a response, the heart could modify the expression of specific cellular signaling. In fact, by proteomics in human biopsies, we observed significant reductions in protein levels of RNA/protein synthesis and metabolic factors, particularly in amino-acid metabolism and mitochondrial respiration. However, the addition of T2DM to the HTN milieu could provoke synergistic effects. T2DM includes additional threats such as obesity, insulin resistance, and overload of glucose and fatty acids that trigger hyperglycemia and dyslipidemia in patients. By proteomics, HTN-T2DM hearts showed higher variations in metabolic, inflammation, RNA/protein synthesis, cytoskeletal, profibrotic, and proapoptotic factors. Among metabolic proteins, most of them were downregulated and linked to mitochondrial respiration and lipid metabolism (Figure 7). In fact, predictive bioinformatics suggested a marked reduction in both fatty acid and glucose oxidation, the TCA cycle, and oxidative phosphorylation, as well as in AMPK and PPARα activation.

Interestingly, by comparing the proteomics patterns of HTN-T2DM versus HTN, most of the differentially expressed proteins were related to mitochondrial respiration and lipid, but not glucose, metabolism. Additionally, HTN-T2DM subjects had class-I obesity and higher levels of LDL-c, TG, and TC than HTN individuals. In fact, the excess of palmitate (HFA), but not high glucose or AngII, reduced the levels of a key transcriptional coactivator and metabolic regulator, PGC1α, in an mTORC1-dependent manner (Figure 8). Interestingly, these actions were enforced by silencing the mTORC2 subunit, Rictor, suggesting that HFA may unbalance the mTOR complexes toward mTORC1 activation. We believe that under HTN and T2DM, some stimuli (i.e., excessive fatty acid) can trigger a cardiac imbalance in mTOR signaling, leading to higher stimulation of the mTORC1 branch and associated deleterious effects. In this sense, mTORC1 activation has been related either to cardiac hypertrophy and depressed autophagy [32] or to the deviation of fatty acid oxidation to glycolysis to confer protection against ischemia-reperfusion injury [33]. As a potential, perhaps temporary, response, the heart could activate adaptive mechanisms which include upregulation of mTORC2 complexes and subsequent mediators (i.e., FoxO1). The link of mTOR complexes and their counter-regulation has been previously suggested [34]. For instance, predominant mTORC1 signaling mediated by suppression of mTORC2 with Rictor increased cardiomyocyte apoptosis and tissue damage after myocardial infarction. However, shifting toward mTORC2 signaling by inhibition of mTORC1 with an mTORC1-regulating subunit, PRAS40, reversed these effects [35]. Intriguingly, specific mTORC1 inhibition by PRAS40, but not by rapamycin, improved cardiac metabolic function, blunted hypertrophic growth, and preserved systolic function in experimental diabetes [36]. Moreover, once mTORC1 is promoted, the downstream signaling towards mitochondria can induce deleterious responses, some of them involving PGC1α. In adipose tissue, the activation of mTORC1 also reduced the PGC1α transcription level and subsequent β-oxidation enzymes [37]. Interestingly, our bioinformatics suggested the reduction of PGC1α only for HTN-T2DM hearts. Thus, we focused on PGC1α-related transcription factors and genes, and we observed that PPARα was also diminished after HFA and recovered with rapamycin. Moreover, a key regulator of fatty acid oxidation in cardiac cells, ACADm, and the subunit of the ETC that catalyzes the oxidation of succinate to fumarate, SDHB, were also downregulated by HFA and improved by rapamycin. In skeletal muscle, a previous report described a reduction of complexes I, II, and IV and that of mitochondrial OCR after mTORC1 activation [38]. In addition, inhibition of mTORC1 by rapamycin increased complex I and improved mitochondrial respiration in liver of lupus-prone mice [39]. Rapamycin also ameliorated diastolic dysfunction by upregulation of complex I and by decrease of both cardiac hypertrophy and passive stiffness in elderly mice [40]. In contrast, inhibition of mTORC1, but not mTORC2, lessened complexes I–IV, mitochondrial respiration, and ATP production in trophoblasts [41]. In this regard, mTORC1 might regulate mitochondrial respiration by targeting other transcription factors (different to PGC1α-PPARα) with opposite actions, depending on environment and cell type and its energetic/nutritional requirements. In addition, pre-incubation with metformin in HFA-stimulated cardiomyocytes mirrored rapamycin effects on mTORC1 inactivation and recovery of PGC1α, ACADm, PPARα, and SDHB levels. HFA in the absence and presence of rapamycin also weakened AMPK activation, which confirms previous reports in myoblasts [42]. Thus, AMPK could respond to HFA and regulate mTORC1 activity.

In addition, the heart needs a high amount of energy to maintain contractile function. Cellular ATP may be originated from multiple energy substrates, mostly fatty acids and carbohydrates (glucose and lactate), and in a minority, ketones and amino acids. However, the contribution of these individual substrates can change depending on substrate availability, hormonal status, and energy demand [43]. In heart failure with or without preserved ejection fraction, cardiac function is reduced, which is accompanied by energy metabolism perturbations and impaired metabolic flexibility [44]. In fact, our HTN-T2DM hearts showed a reduction in fatty acid oxidation enzymes and downstream pathways (TCA, mitochondrial electron chain), but also in glycolysis. Additionally, HFA-stimulated cardiomyocytes decreased β-oxidation and ATP formation from mitochondrial respiration. Interestingly, these cardiomyocytes did not switch to glycolysis either, suggesting metabolic inflexibility also under HFA (at least at 90 uM for 24 h).

Altogether, in cardiomyocytes, mitochondrial alterations derived from fatty acid overload might be attenuated by mTORC1 inhibitors or by AMPK and PGC1α-PPARα activators. These molecular signalings could be proposed as therapeutic targets for hearts under HTN and/or T2DM. In fact, the activator of PGC1α, ZLN005, was able to increase key β-oxidation enzymes and ETC complexes under HFA incubation, which could potentially enhance ATP production and metabolic flexibility. Further experiments in vivo will be needed to confirm this hypothesis. Additionally, the relationships between cardiac mTORC1, PGC1α, and mitochondrial regulators under different availability of nutrients, O_2_, and ATP demand are still unknown.

## 4. Materials and Methods

### 4.1. Population Study

Intraoperative cardiac biopsies from the interventricular septum were obtained from patients with HTN with/without concomitant T2DM who underwent coronary artery bypass grafting (CABG) at Doce Octubre and Fundación Jiménez Díaz hospitals (Madrid). Biopsies were taken from the myocardial area without scarring. Following European guidelines, HTN was defined as systolic blood pressure ≥ 140 mmHg and/or diastolic blood pressure ≥ 90 mmHg. T2DM was diagnosed when glycosylated hemoglobin (Hb1Ac) ≥ 6.5% (48 mmol/mol) and fasting plasma glucose ≥ 7.0 mmol/L. If inconclusive, it was diagnosed when 2h-plasma glucose after an oral glucose tolerance test (OGTT) ≥ 11.1 mmol/L.

The exclusion criteria included (i) the presence of myocardial scar evident in echocardiogram and/or electrocardiogram (except lateral scar), (ii) systemic disease or (iii) any chronic treatment except those against atherothrombosis or its risk factors, and (iv) non-Caucasian race. The control group included patients without HTN or T2DM undergoing cardiac valve replacement or CABG. All patients received pharmacological treatment including anti-hypertensive, anti-hyperlipidemic, anti-thrombotic, and/or anti-diabetic drugs. Acronyms used include: ACEi: Angiotensin converting enzyme inhibitors, ARB: Angiotensin II receptor blockers, MRA: Mineralocorticoids receptor antagonists, Antithrombotic (Acenocumarol: Sintrom), DAPT: Dual antiplatelet therapy (aspirin clopidogrel, ticagrelor or prasugrel) (Table 1C). The investigation fulfilled the principles contained in the Declaration of Helsinki and subsequent reviews, as well as the prevailing Spanish legislation on clinical research in human subjects. The study protocol was approved by the Ethics Committee of Clinical Investigation of Doce Octubre and Fundación Jiménez Díaz Hospitals (Ref.: PIC 02/2010).

### 4.2. Proteomics Approach

#### 4.2.1. Sample Preparation

A piece of each biopsy (20–30 mg) from the control (N = 5), HTN (N = 7), and HTN-T2DM (N = 7) patients was incubated in lysis buffer containing 7 M urea, 2 M Thiourea, 4% CHAPS, and 5 mM DTT to extract total proteins with a tissue homogenizer (Precellys, Bertin Technologies). Then, proteins were digested following the filter-aided sample preparation (FASP) method described by Wisniewski et al. [45] with minor modifications. Trypsin was added at a trypsin: protein ratio of 1:50, and the mixture was incubated overnight at 37 °C, dried out in an RVC2-25 speed-vac concentrator (Christ), and resuspended in 0.1% formic acid. Peptides were desalted and resuspended in 0.1% formic acid using C18 stage tips (Millipore, Burlington, MA, USA).

#### 4.2.2. Mass Spectrometry Analysis

Samples were analyzed in a novel hybrid trapped ion mobility spectrometry-quadrupole time of flight mass spectrometer (timsTOF Pro with PASEF, Bruker Daltonics, Billerica, MA, USA) coupled online to a Nano Elute liquid chromatograph (Bruker). Samples (200 ng) were directly loaded in a 15 cm Bruker Nano elute FIFTEEN C18 analytical column (Bruker) and resolved at 400 nL/min in 3–40% acetonitrile in a 100 min gradient. The column was heated to 50 °C using an oven.

#### 4.2.3. Protein Identification and Quantification

Protein identification and quantification were carried out using PEAKS software v.8.5 (Bioinformatics solutions). Searches were carried out against a database consisting of human entries (Uniprot/Swissprot), with precursor and fragment tolerances of 20 ppm and 0.05 Da. Only proteins identified with at least two peptides at FDR < 1% were considered for further analysis. Data were loaded onto the Perseus platform v.2.0.9.0 [46] and further processed (log2 transformation, imputation). A *t*-test was applied to determine the statistical significance of the differences detected.

#### 4.2.4. Bioinformatic Predictive Analysis

The Ingenuity pathway analysis (IPA^®^, 2000-2019 QIAGEN) was used for a characterization of the molecular events lying behind the differential protein patterns under examination. The calculated *p*-values for the different analyses performed determined the probability that the association between proteins in the dataset and a given process, pathway, or upstream regulator is explained by chance alone, based on a Fisher’s exact test (*p*-value < 0.05 being considered significant). The activation z-score represents the bias in gene regulation that predicts whether the upstream regulator exists in an activated (positive values) or inactivated (negative values) state based on the knowledge of the relation between the effectors and their target molecules. Only values with |Z-score| ≥ 2 were considered. With respect to the graphical representation of the data, a red/green color code was used for proteins detected in the proteomics experiment; red proteins being downregulated, and green proteins upregulated. IPA correlated the expression pattern of the proteins under analysis with the information contained in its knowledgebase to predict the state of activation/inhibition of these upstream regulators/biological processes.

### 4.3. In Vitro Approach

#### 4.3.1. Cultured Cardiomyocytes and Cell Stimuli

Cardiomyocyte H9c2(2-1) is a subclone of the original clonal cell line derived from embryonic BD1X rat heart (ventricle) tissue (ATCC, Manassas, VA, USA). H9c2(2-1) cells were obtained from ATCC and cultured in Dulbecco’s modified Eagle’s medium (DMEM) supplemented with 10% fetal bovine serum (FBS) and maintained at 37 °C in a humidified atmosphere of 5% CO_2_ and 95% air. Cardiomyocytes were fed every two days and subcultured once they reached 80% confluence in a 1:3 split. Cells were seeded at 1 × 10^4^ cells/cm^2^ and incubated with major components of HTN and T2DM–obese milieu. The pro-HTN condition was simulated by incubation with Angiotensin II (1 × 10^−9^ mol/L). Hyperlipidemia was mimicked by adding a high fatty acid concentration (HFA; 90 µM Na^+^-palmitate) conjugated with BSA in a 3:1 ratio, and hyperglycemia was mirrored by incubating cells with a high glucose concentration (HG; 25 mM D-glucose), as previously published [47]. All stimuli were incubated for 24h after overnight starvation in non-FBS DMEM. In addition, rapamycin (10 nM; ThermoFisher Scientific, Waltham, MA, USA) or metformin (5 mM; Merck, Rahway, NJ, USA) as a specific mTORC1 inhibitor or AMPK activator, respectively, were added one hour before the treatments. Similarly, SR-18292 (MedChem Express, South Brunswick, NJ, USA), as a PGC-1α inhibitor, which increases PGC-1α acetylation, suppresses gluconeogenic gene expression, and reduces glucose production in hepatocytes, was added at 10 µM to the cells [48]. Additionally, ZLN005 (SML0802, Sigma-Aldrich, Schnelldorf, Germany), an activator of PGC-1α, which also stimulates PGC-1α expression and downstream genes by activating AMPK, was used at 5–10 μg/mL [49]. MHY1485 (1 mM; ThermoFisher Scientific), as an activator of the mTOR common subunit for mTORC1 and mTORC2 complexes, was added together with the stimuli. At these conditions, all stimuli did not induce toxic or lethal effects on cardiomyocytes.

#### 4.3.2. Gene Silencing

Small interfering (si) RNA sequences specific for rat Rictor and PGC1α genes were purchased from ThermoFisher Scientific. Then, cardiomyocytes were seeded at 70% confluence before adding the siRNA (siRictor or siPGC1α) carried by Lipofectamine^®^ 3000 according to the manufacturer’s instructions (Invitrogen, Waltham, MA, USA). After 24 h of transfection, siRNA was removed, and cells were washed with PBS. Then, they were incubated with HFA as described above. The protein expression for Rictor and PGC1α was reduced by at least 65% in both cases for all stimulations (Appendix A). Similar data were found for the related mRNA expression.

#### 4.3.3. Protein Determination

From cultured cardiomyocytes, proteins were extracted by adding lysis buffer (Tris 50 mM, NaCl 0.15 M, EDTA 2.5 mM, TRITON X-100 0.2%, IGEPAL 0.3%) to stimulated cells. The protein concentration was estimated by the Bicinchoninic acid assay (BCA Protein Assay Kit, ref 23225, ThermoFisher Scientific). Then, 30–50 ug per sample were loaded on SDS-PAGE gels and separated by electrophoresis. Proteins were then transferred onto nitrocellulose membranes (Bio-Rad, Hercules, CA, USA). The unspecific sites were blocked with skimmed milk in 5% TBST. Primary antibodies were used to quantify the protein levels. Anti-ACADM (ref.: PA5-27201), -PGC-1 alpha (PA5-72948), and -GAPDH (MA5-15738) antibodies were purchased from ThermoFisher Scientific. Anti-Rictor (SAB4200141) was from Sigma and anti-Phospho-AMPKα (Thr172) monoclonal (2535), -AMPKα (2532), -P70S6 Kinase (34475S), and -Phospho-P70S6 (9204S) were from Cell Signaling. The antibody–antigen binding was detected by chemiluminescence. Secondaries antibodies were goat anti-Mouse IgG (H + L) (ref: G21040), goat anti-Rabbit IgG (H + L) (ref: G21234), and goat anti-mouse (31430) from ThermoFisher Scientific. Then, antibody complexes were quantified by the ImageJ software v.1.53. All experiments were replicated at least four times.

#### 4.3.4. Quantitative Gene Expression

Total RNA was extracted from cardiomyocytes by using Trizol reagent (ThermoFisher Scientific). Then, after spectrophotometric quantification (Nanophotometer^®^ N60, IMPLEN), cDNA synthesis and real-time quantitative PCR analysis (qPCR) were performed as previously described [47]. In brief, reverse transcription with the High-Capacity cDNA Reverse Transcription Kit (ref: 4368813) (Applied biosystems, Waltham, MA, USA) of the extracted RNA to cDNA was performed in Veriti Thermal Cycler (ThermoFisher Scientific). Then, cDNA, the universal qPCR master mix (TaqMan™ Universal PCR Master Mix ref:4318157, Applied biosystems), and the specific gene expression assays (Fam-fluorophores) for PPARα (Rn00566193_m1) or SDHB (Rn01515728_m1) were mixed in a final volume of 10 µL (Applied Biosystems). The qPCR was performed in the StepOnePlus™ Real-Time PCR System (ThermoFisher Scientific). Each sample was run in triplicate and internal variations higher than 0.3 cycles were not considered. The relative gene expression was revealed by the comparative ΔΔC_T_; method (StepOne™ Plus v.2.3 software). As a housekeeping gene, we used the eukaryotic ribosomal 18s labeled with VIC-fluorophore (Hs99999901_s1; ThermoFisher Scientific).

#### 4.3.5. ATP Quantification

The cellular ATP levels were quantified by a bioluminescence assay Molecular Probes (ATP Determination Kit, ref.: A22066, Invitrogen) based on luciferase’s requirement for ATP production (Abs ~560 nm at pH 7.8). Cardiomyocytes were incubated with HFA and/or siPGC-1α or siRictor. Then, ATP was determined following the manufacturer’s instructions. For normalization, cells were lysed with ATP-releasing buffer (100 mM KH_2_PO_4_, 2 mM EDTA, 1 mM dithiothreitol, and 1% Triton X-100 at pH 7.8) and total proteins were quantified by the Bicinchoninic acid (BCA) protein assay [50]. The experiments were replicated at least three times.

#### 4.3.6. Mitochondrial Bioenergetic Response

A Seahorse XF Pro Analyzer (Agilent technologies, Santa Clara, CA, USA) was used to measure cell respiration in cardiomyocytes. We seeded 1 × 10^4^ cells/well into XF Pro M microplates (Cat. No: 103774-100). Cardiomyocytes were incubated with HFA (90 μM for 24 h) in the absence or presence of inhibitors or silencers, as commented above. Then, on the day of the assay, the medium was replaced with XF DMEM supplemented with 1 mM sodium pyruvate, 2 mM glutamine, and 1 mM D-glucose (Cat. No.: 103575–100). Cells were cultured at 37 °C in a non-CO_2_ incubator for an additional hour and refreshed with XF DMEM supplemented before measurement. The real-time ATP rate assay (Cat. No: 103592-100) was used to simultaneously detect mitochondrial and glycolytic ATP production and the oxygen consumption rate (OCR). Then, 1.5 μM oligomycin (an ATP synthase (complex V) inhibitor) and 0.5 μM rotenone/antimycin A (inhibitors of complex I and III, respectively) were automatically injected into the wells. Oligomycin induces a reduction in oxygen consumption associated with ATP synthesis, while rotenone and antimycin A reduce the flux of electrons in the electron transport chain (ETC). The ATP production rate and OCR were normalized by protein concentration (µg/µL) by using the BCA Protein Assay Kit (Cat. No: 23225; Pierce™). Finally, data analysis was performed with the seahorse Wave Pro & XF Pro-Controller Software v.2.6 (Agilent technologies, Santa Clara, CA, USA), following a previous report [51]. The experiments were replicated 4–8 times.

### 4.4. Statistical Analysis

For proteomic analysis, variables with normal distribution were expressed as mean values and standard deviation. The student’s *t*-test was used to compare average relationships between groups, and *p* < 0.05 was considered significant. For pharmacological treatments, qualitative variables were presented as relative frequencies (percentages). Associations between qualitative variables were examined using Fisher’s exact test, followed by Bonferroni post-hoc analysis for multiple comparisons. Significant associations were indicated as *p* < 0.05. For the vitro experiments, the normality of the quantitative variables was analyzed by the Shapiro–Wilk test. Non-parametric variables were analyzed by using the Kruskal–Wallis test, following Dunn’s multiple comparisons test. Variables with normal distribution were studied by one-way ANOVA followed by a Tukey’s test. The results were expressed as median (interquartile range) or mean (standard deviation), and *p* < 0.05 was considered significant. Statistical analyses were performed by using the statistical package for social science (SPSS, IBM, Armonk, NY, USA) v.26.0 and the Prism version 9 GraphPad Software, San Diego, CA, USA.

## 5. Study Limitation

A group of patients with only T2DM but not HTN could have been of interest to better analyze, by proteomics, the unique influence of T2DM in cardiac failure. However, the high prevalence of comorbidities in T2DM subjects impeded this recruitment. In this sense, the influence of obesity should be similarly examined. In addition, a larger size of intraoperative cardiac biopsies would have allowed further studies to confirm the implication of the mTORC1-PGC1α-PPARα axis and linked mediators in human hearts. On the other hand, all patients received multiple pharmacological treatments that might have affected the proteomics and bioinformatics results. Nevertheless, though hyperglycemia was attenuated in HTN-T2DM subjects, as expected, HTN and dyslipidemia were partially uncontrolled in both HTN and HTN-T2DM individuals.

## 6. Conclusions

Both HTN and T2DM can damage cardiac physiology and function in patients with coronary heart disease, and these effects could be extended when both pathologies are concomitant, which is the most prevalent situation. We discovered in HTN-T2DM hearts a greater number and degree of alterations in protein content. Importantly, even in the early stages of coronary heart disease, when systolic function is preserved, some molecular mechanisms can be activated. A general downregulation of metabolic factors, most of them linked to mitochondrial respiration and lipid metabolism, may be relevant. The lipid component could be crucial for cardiac responses in HTN-T2DM hearts (Figure 8). Indeed, the excess of fatty acid-activated mTORC1 complexes in cardiomyocytes consequently reduced the PGC1α-PPARα signaling. Some PGC1α-related genes involved in fatty acid oxidation (e.g., ACADm) and in the ETC (e.g., SDHB) were also lessened, which probably impaired mitochondrial/glycolytic ATP production. Thus, the mTORC1-PGC1α-PPARα axis might be targeted for future therapies to prevent cardiac dysfunction associated with HTN and/or T2DM. Further experiments in vivo with mTORC1 inhibitors and mTORC2 or PGC1α-PPARα activators might lead to promising results.

## Figures and Tables

**Figure 1 ijms-24-08629-f001:**
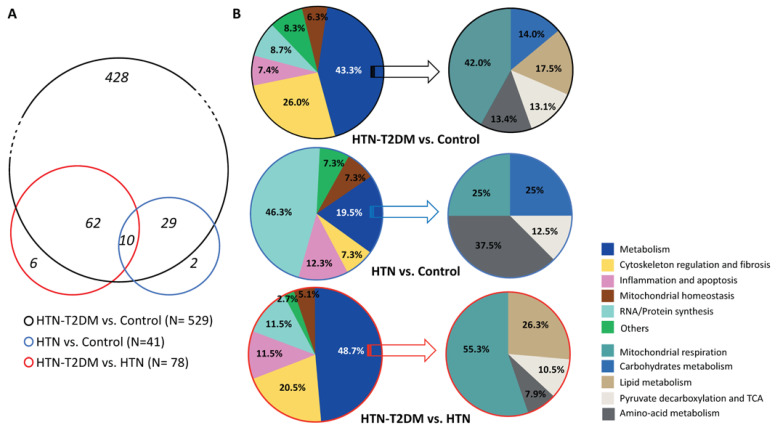
Cardiac proteomic patterns under HTN and HTN-T2DM. (**A**) The number of significantly altered proteins (*p* < 0.05) found in HTN-T2DM vs. control (black circle), HTN vs. control (blue circle), and HTN-T2DM vs. HTN (red circle). (**B**, **left**) The distribution of changed proteins in relation to metabolism, cytoskeleton regulation, inflammation and apoptosis, mitochondrial physiology, RNA/Protein synthesis, and others of the three comparative groups (black, blue, and red). (**B**, **right**) The distribution of metabolic proteins clustered by mitochondrial respiration, carbohydrates metabolism, lipid metabolism, pyruvate decarboxylation and TCA, and amino-acid metabolism of the three comparative groups (black, blue, and red). HTN-T2DM (*n* = 7), HTN (*n* = 7), and Control (*n* = 5). The student’s *t*-test was used to compare the mean of measurements between both groups.

**Figure 2 ijms-24-08629-f002:**
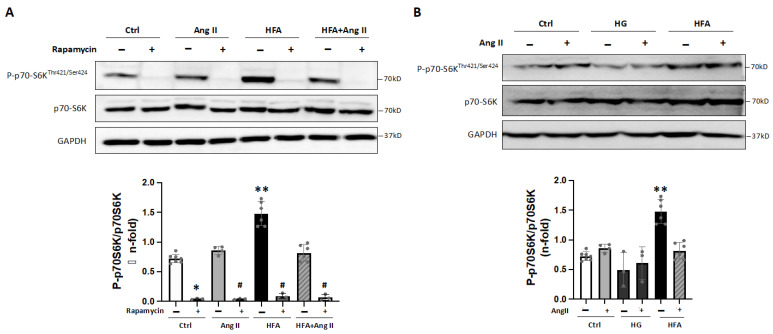
High fatty acids as a major activator of mTORC1 in cardiomyocytes. (**A**) Protein expression of Phospho-p70-S6K^Thr421/Ser424^ and p70-S6K under Angiotensin II and/or HFA stimulation for 24 h. Some cells were also preincubated with rapamycin. GAPDH was used as a protein loading control. * *p* < 0.05 and ** *p* < 0.01 vs. control, # *p* < 0.05 vs. AngII, HFA or HFA + AngII. (**B**) Expression of Phospho-p70-S6K^Thr421/Ser424^ and p70-S6K after high D-glucose (HG) or HFA and/or Angiotensin II. ** *p* < 0.01 vs. control. The data distribution was analyzed by the Shapiro–Wilk test. The non-parametric variables were studied by using the Kruskal–Wallis test, following Dunn’s multiple comparisons test, and *p* < 0.05 was considered significant.

**Figure 3 ijms-24-08629-f003:**
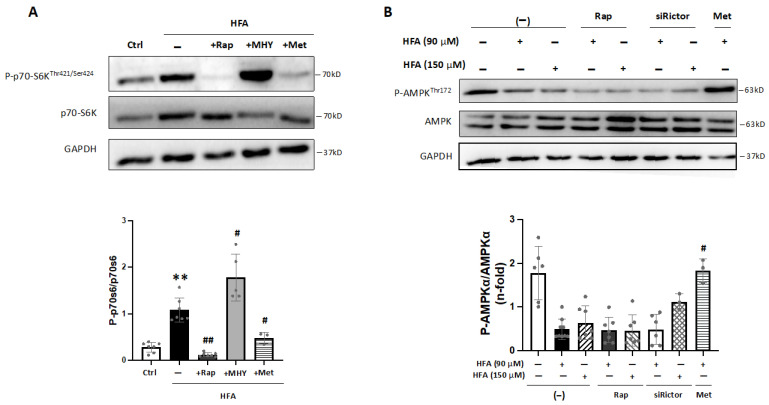
Role of AMPK in the HFA-induced mTORC1 activation. (**A**) Protein levels of Phospho-p70-S6K^Thr421/Ser424^ and p70-S6K after HFA incubation in cardiomyocytes. Some cells were also pre-treated with rapamycin, MHY1485, or metformin. GAPDH was used as a protein loading control. ** *p* < 0.01 vs. control, # *p* < 0.05 and ## *p* < 0.01 vs. HFA. (**B**) Protein levels of Phospho-AMPK^Thr172^ and AMPK after HFA incubation (90 and 150 μM). Some cells were preincubated with rapamycin, siRictor, or metformin. # *p* < 0.05 vs. HFA (90 μM). The data distribution was studied by the Shapiro–Wilk test, and non-parametric variables were analyzed by using the Kruskal–Wallis test, following Dunn’s multiple comparisons test. *p* < 0.05 was considered significant.

**Figure 4 ijms-24-08629-f004:**
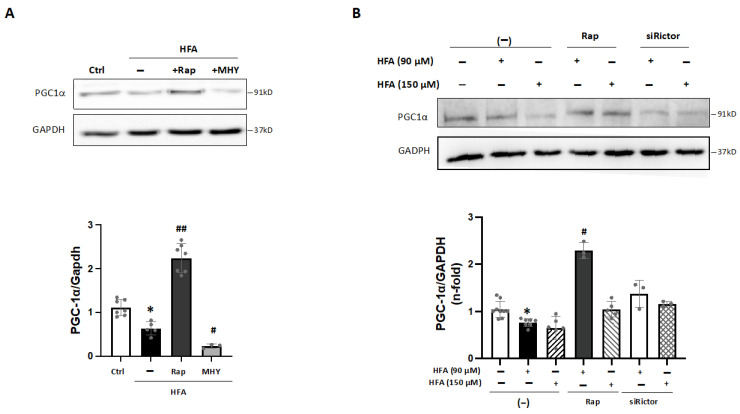
Regulation of PGC1α under HFA-mTORC1 stimulation in cardiomyocytes. (**A**) Expression of PGC1α in cardiomyocytes after HFA and/or preincubation with rapamycin or MHY1485. * *p* < 0.05 vs. control, # *p* < 0.05 and ## *p* < 0.01 vs. HFA. (**B**) Expression of PGC1α after HFA (90–150 μM) with/without preincubation with rapamycin or siRictor. * *p* < 0.05 vs. control, # *p* < 0.05 vs. HFA. The data distribution was analyzed by the Shapiro–Wilk test. The non-parametric variables were studied by using the Kruskal–Wallis test, following Dunn’s multiple comparisons test, and *p* < 0.05 was considered significant.

**Figure 5 ijms-24-08629-f005:**
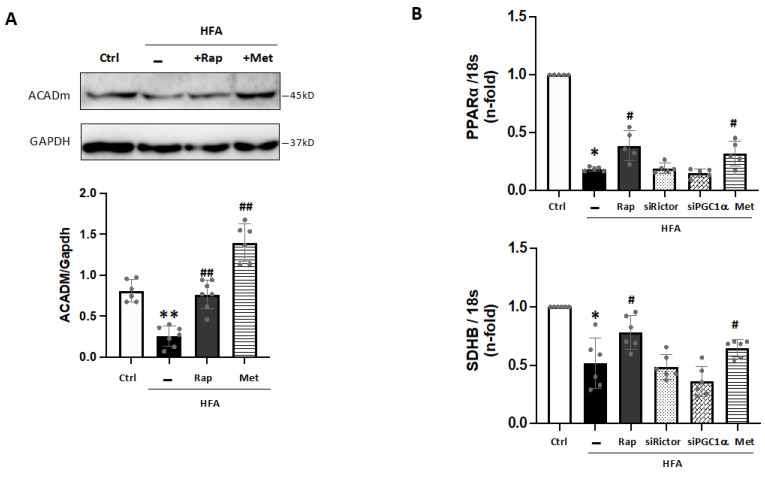
Downregulation of PGC1α-related factors under HFA. (**A**) Expression of ACADm in cardiomyocytes stimulated with HFA and/or rapamycin or metformin preincubation. ** *p* < 0.01 vs. control, ## *p* < 0.01 vs. HFA. (**B**) Gene expression of PPARα and SDHB after HFA with/without pre-treatment with rapamycin, siRictor, siPGC1α, or metformin. * *p* < 0.05 vs. control, # *p* < 0.05 vs. HFA. The data distribution was examined by the Shapiro–Wilk test. The non-parametric variables were studied by using the Kruskal–Wallis test, following Dunn’s multiple comparisons test. *p* < 0.05 was considered significant.

**Figure 6 ijms-24-08629-f006:**
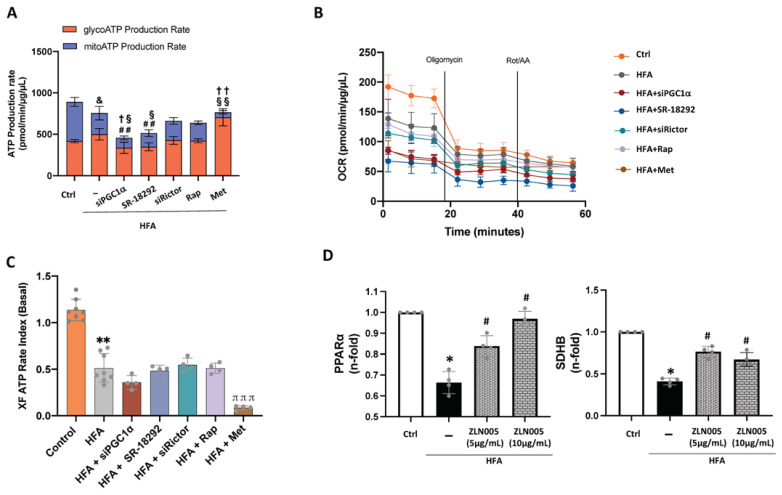
Glycolytic and mitochondrial ATP formation under HFA-mTORC1 activation in cardiomyocytes. Bioenergetic quantification of HFA-stimulated cardiomyocytes. The Seahorse XF Pro Analyzer equipment was used to quantify (**A**) total, glycolytic (in red), and mitochondrial (in blue) ATP synthesis, (**B**) the oxygen consumption rate (OCR), and (**C**) the XF ATP rate index as the ratio between mitochondrial ATP production and glycolytic ATP synthesis in HFA-stimulated cells with/without silencing of PGC1α or Rictor, or specific inhibitors for PGC1α (SR-18292) and mTORC1 (rapamycin) or AMPK activator (metformin). ^##^
*p* < 0.01 vs. total ATP-HFA, ^&^
*p*< 0.05 vs. mitochondrial ATP control, ^§^
*p*< 0.05 vs. glycolytic ATP HFA, ^§§^
*p*< 0.01 vs. glycolytic ATP HFA, ^†^
*p* < 0.05 vs. mitochondrial ATP-HFA, and ^††^
*p* < 0.01 vs. mitochondrial ATP-HFA. Additionally, ** *p* < 0.01 vs. control and ^πππ^
*p* < 0.001 vs. HFA. (**D**) Gene expression of PPARα and SDHB after HFA with/without ZLN005 pre-treatment at 5 and 10 μg/mL. * *p* < 0.05 vs. control, # *p* < 0.05 vs. HFA. The data distribution was studied by the Shapiro–Wilk test. Variables with normal distribution were analyzed by one-way ANOVA followed by Tukey’s test, while the non-parametric variables were studied using the Kruskal–Wallis test, following Dunn’s multiple comparisons test. *p* < 0.05 was considered significant.

**Figure 7 ijms-24-08629-f007:**
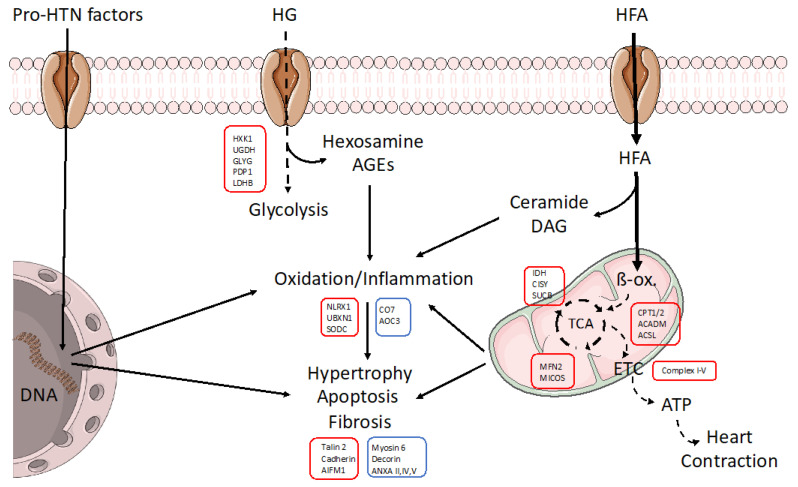
Schematic model of the molecular alterations induced in cardiac cells under HTN-T2DM. In an HTN-T2DM scenario, the stimulation of pro-hypertensive factors such as RAA peptides, sympathetic neurotransmitters, and oxidative/inflammatory factors induces the expression of pro-oxidant/-inflammatory/-hypertrophic proteins. In addition, the lack of insulin sensitivity reduces glucose utilization and the overload of fatty acids as a unique energetic substrate. HFAs saturate β-oxidation and deviate to fatty acid metabolites (ceramide, diacylglycerol; DAG) with high pro-oxidant and apoptotic properties. The non-assimilated glucose also leads to pro-oxidant molecules such as hexosamine and AGEs. All these effects can damage mitochondria and energy supply for contraction and could be amplified under obesity. Interestingly, by proteomics, we found key proteins that could be targeted for cardiac dysfunction in these patients. In red boxes, those factors that were downregulated. In blue, those proteins that were upregulated. ETC: electron transport chain, ACSL: Long-chain fatty acid–CoA ligase, AIFM: Apoptosis-inducing factor 1, ANXA: Annexin, CISY: Citrate synthase, AOC3: Amine Oxidase Copper Containing 3, CO7: Complement component C7, CPT: Carnitine O-palmitoyl transferase, GLYG: Glycogenin-1, HXK1: Hexokinase-1, IDH: Isocitrate dehydrogenase, LDHB: L-lactate dehydrogenase B chain, MFN2: Mitofusin-2, NLRX1: NLR family member X1, PDP1: Pyruvate dehydrogenase [acetyl-transferring]-phosphatase 1, SODC: Superoxide dismutase [Cu-Zn], SUCB: Succinyl-CoA ligase [ADP-forming] subunit beta, UBXN1: UBX domain-containing protein 1, UGDH: UDP-glucose 6-dehydrogenase.

**Figure 8 ijms-24-08629-f008:**
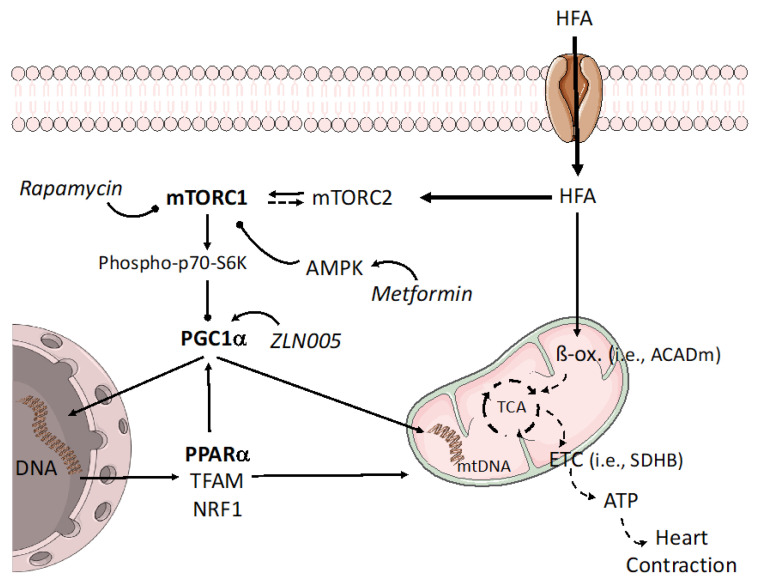
Potential regulation of the mTORC1-PGC1α-PPARα axis under overloaded fatty acid. In addition to saturating β-oxidation, the excess fatty acid can also shift the balance of mTOR complexes toward mTORC1, which is able to block PGC1α levels and related gene expression. Among them, PPARα and other mitochondrial regulators (i.e., TFAM, NRF1) may be attenuated at the nuclei and mitochondrion, worsening mitochondrial function. mTORC1 inhibitors and AMPK activators, such as rapamycin or metformin, respectively, might mitigate these actions. In this sense, PGC1α enhancers could improve mitochondrial respiration and fatty acid utilization genes. mtDNA: mitochondrial DNA, TFAM: mitochondrial transcription factor A, NRF1: Nuclear respiratory factor 1.

**Table 1 ijms-24-08629-t001:** Characterization of the population. (**A**) Control (*n* = 5), HTN (*n* = 7), and HTN-T2DM (*n* = 7) patients with coronary heart disease and aortic/mitral valvopathy or acute myocardial infarction (AMI) with ST- or non-ST-elevation were intervened on for valve replacement or coronary artery bypass grafting (CABG). At this time, the LVEF and LVDD and the presence of LVH were quantified. The results are shown as mean ± SD or percentage. The student’s *t*-test was used to compare the mean of measurements between both groups. The non-pathological ranges for LVEF and LVDD are depicted according to the guidelines of the European Association of Cardiovascular Imaging [14]. (**B**) The anthropometric parameters were quantified (BMI, body mass index). Additionally, the rate of controlled hypertension, the levels of Hb1Ac, and the lipid profile (LDL-c, HDL-c, triglycerides, and total cholesterol) were quantified. The results are shown as mean ± SD or percentage. The student’s *t*-test was used to compare the mean of measurements between both groups. The non-pathological ranges for Hb1Ac and the lipid profile are shown according to the guidelines of the European Society of Cardiology [15,16]. ^†^ Level of LDL-c for patients with high-risk of cardiovascular disease [presence of atherosclerotic disease, including previous acute coronary syndrome (AMI or unstable angina); stable angina; coronary revascularization (percutaneous coronary intervention, CABG, and other arterial revascularization procedures); stroke and transient ischemic attack; peripheral arterial disease]. * *p* < 0.05 vs. control. (**C**) The pharmacological treatments were included as percentages (%) and analyzed using Fisher’s exact test followed by Bonferroni post-hoc test. * *p* < 0.05 vs. control and ^#^
*p* < 0.05 vs. HTN. ACEi: Angiotensin converting enzyme inhibitors, ARB: Angiotensin II receptor blockers, MRA: Mineralocorticoids receptor antagonists, Antithrombotic (Acenocumarol: Sintrom), DAPT: Dual antiplatelet therapy (aspirin clopidogrel, ticagrelor or prasugrel).

**(A)**
	**Control**(Mean ± SD or %)***n* = 5**	**HTN** **(Mean ± SD or %)** ***n* = 7**	**HTN-T2DM** **(Mean ± SD or %)** ** *n* ** ** = 7**	**Non-Pathological** **Ranges**
**Cardiac injury**	Aortic/mitral valvopathy	AMI (ST/non-ST-elevation)	AMI (ST/non-ST-elevation)	-
**Cardiac surgery**	Valve repl./CABG	CABG	CABG	
**LVEF (%)**	57.0 ± 6.7	56.9 ± 5.1	59.1 ± 5.7	52–72 (male)54–74 (female)
**Presence of LVH (%)**	40.0	14.2	28.5	-
**LVDD (mm)**	45.6 ± 9.9	42.1 ± 4.5	44.0 ± 3.2	42–58 (male)38–52 (female)
**(B)**
	**Control** **(Mean ± SD or %)** ***n* = 5**	**HTN** **(Mean ± SD or %)** ***n* = 7**	**HTN-T2DM** **(Mean ± SD or %)** ** *n* ** ** = 7**	**Non-Pathological** **Ranges**
**Age (years-old)**	71.2 ± 8.4	64.7 ± 7.7	72.5 ± 8.8	-
**Male sex (%)**	40.0	85.7	85.7	-
**BMI (kg/m^2^)**	29.5 ± 4.8	29.4 ± 6.6	32.8 ± 5.3	≤30 kg/m^2^
**Controlled HTN (%)**	100	71.4	22.2	-
**HbA_1c_ (%)**	<5.7	5.8 ± 0.1	6.9 ± 1.0 *	<7
**Lipid Profile**				
**LDL-c (mg/dL)**	77.4 ± 29.4	83.1 ± 12.2	94.4 ± 30.3	≤100 or ≤55 ^†^
**HDL-c (mg/dL)**	52.6 ± 16.3	37.4 ± 5.13	38.8 ± 7.2	>40
**TG (mg/dL)**	94.4 ± 8.8	145.6 ± 56.1	172.0 ± 60.9 *	<150
**TC (mg/dL)**	148 ± 28.6	149.3 ± 20.7	176.5 ± 25.7	<155
**(C)**
		**Control (%)** ** *n* ** ** = 5**	**HTN (%)** ** *n* ** ** = 7**	**HTN-T2DM (%)** ** *n* ** ** = 7**
**HTN**				
	**ACEi**	60.0	57.1	57.1
**ARB**	20.0	14.3	14.3
	**MRA**	40.0	28.60	14.3
	**β-blocker**	80.0	100.00	57.1
**T2DM**				
	**Insulin**	0.0	0.00	57.1
**Metformin**	0.0	0.00	71.4 *^#^
**Hyperlipidemia**				
	**Simvastatin**	40.0	0.00	57.1
**Atorvastatin**	0.0	100.0 *	42.9
**Thrombosis**				
	**Aspirin**	0.0	57.1	71.4 *
**DAPT**	20.0	42.9	28.6
**Sintrom**	80.0	0.0 *	0.0 *

**Table 2 ijms-24-08629-t002:** Candidate pathways predicted for HTN-T2DM and HTN hearts.

Candidate Pathways (IPA^**®**^)	HTN vs. Ctrl. (z-Score)	HTN-T2DM vs. Ctrl. (z-Score)			
Sirtuin Signaling Pathway	—	3.371			
Acute Phase Response Signaling	—	2.683			
Ketogenesis	—	−2			
Fatty Acid α-oxidation	—	−2			
Ketolysis	—	−2.236			
Acetyl-CoA Biosynthesis I (PDH Complex)	—	−2.236			
AMPK Signaling	—	−2.887			
Glycolysis	—	−3.317			
Gluconeogenesis	—	−3.317			z-score ≥ 5
eIF2 Signaling	−2.449	−3.153			2 ≤ z-score < 5
Fatty Acid β-oxidation	—	−3.742			−5 < z-score ≤ −2
TCA Cycle	—	−4.123			z-score ≤ −5
Oxidative Phosphorylation	—	−8.062		—	−2 < z-score < 2; *p* > 0.05

Candidate pathways predicted for HTN-T2DM and HTN hearts. After proteomics, the list of differentially expressed proteins against control in HTN-T2DM (*n* = 529) and HTN (*n* = 41) samples were analyzed by IPA^®^. The candidate molecular pathways are depicted for HTN-T2DM and HTN hearts as bias-corrected activation z-scores. A significant activation or inhibition was considered if z-score ≥ 2 (light green–dark green) or z-score ≤ −2 (orange–red), respectively. The non-significant z-scores are in white color with a dash. PDH: Pyruvate dehydrogenase, AMPK: 5′ AMP-activated protein kinase, eIF2: Eukaryotic Initiation Factor 2, TCA: Tricarboxylic acid cycle. The calculated *p*-values were based on Fisher’s exact test (*p*-value < 0.05 was considered significant).

**Table 3 ijms-24-08629-t003:** Upstream regulators proposed for HTN-T2DM and HTN hearts.

Upstream Regulators (IPA^**®**^)	HTN vs. Ctrl. (z-Score)	HTN-T2DM vs. Ctrl. (z-Score)	Main Cellular Function			
RICTOR	3	8.752	Subunit of mTORC2. Regulation of cellular processes			
MAP4K4	—	6.164	Anti-oxidation			
CD 437	2	5.425	Pro-apoptosis			
NUPR1	2	—	Anti-oxidation			
PPARγ	—	−3.168	Regulation of lipid metabolism			
CD28	−2	−3.965	Anti-inflammation			
NFE2L2	—	−4.201	Anti-oxidation			
PPARα	—	−4.423	Regulation of lipid metabolism			
CD3	−2	−4.876	Anti-inflammation			z-score ≥ 5
MYCN	−2.985	−5.565	Cardiomyocyte proliferation			2 ≤ z-score < 5
PGC1α	—	−6.069	Mitochondrial biogenesis metabolism			−5 < z-score ≤ −2
INSR	—	−6.215	Regulation of glucose lipid metabolism			z-score ≤ −5
MYC	−2.701	−6.216	Regulation of glucose metabolism mitochondrial biogenesis		—	−2 < z-score < 2; *p* > 0.05

Upstream regulators proposed for HTN-T2DM and HTN hearts. After proteomics, the list of differentially expressed proteins against control in HTN-T2DM (*n* = 529) and HTN (*n* = 41) myocardia were analyzed by the IPA^®^, and potential upstream regulators were suggested. A significant activation or inhibition was considered if z-score ≥ 2 (light green–dark green) or z-score ≤ −2 (orange–red), respectively. The non-significant z-scores are in white color with a dash. The main cellular functions were described by the Uniprot database (www.uniprot.org; accessed in October 2022). RICTOR: Rapamycin-insensitive companion of mammalian target of rapamycin, MAP4K4: Mitogen-activated protein kinase kinase kinase kinase-4, CD437: Synthetic retinoid 6-[3-(1-adamantyl)-4-hydroxyphenyl]-2-naphthalene carboxylic acid, NUPR1: Nuclear protein-1, PPARγ: Peroxisome proliferator-activated receptor-gamma, CD28: Cluster of Differentiation 28, NFE2L2: Nuclear factor erythroid 2-related factor 2, PPARα: Peroxisome proliferator-activated receptor alpha, CD3: Cluster of differentiation 3, MYCN: N-myc proto-oncogene protein, PGC1α: Peroxisome proliferator-activated receptor gamma coactivator 1-alpha, INSR: insulin receptor, MYC: bHLH transcription factor. The calculated *p*-values were based on Fisher’s exact test (*p*-value < 0.05 was considered significant).

## Data Availability

No new data were created.

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
