# Peer review of "Potential Role of the mTORC1-PGC1α-PPARα Axis under Type-II Diabetes and Hypertension in the Human Heart"

_ijms, 2023, doi:10.3390/ijms24108629_

Round 1

Reviewer 1 Report

The manuscript by Hang et al. et al. describes with type-2 diabetes (T2DM) and arterial hypertension (HTN) are the significant risk factors in Heart failure. Proteomics and bioinformatics of cardiac biopsies from patients with or without T2DM/HTN revealed substantial changes in 677 proteins. HTN-T2DM had 529, and HTN had 41 proteins altered versus control, with 81% of HTN-T2DM proteins distinct from HTN. Bioinformatics implicated mTOR signaling and reduced activation of AMPK/PPARα, with PGC1α, fatty acid oxidation, and oxidative phosphorylation regulation. In vitro studies using cultured cardiomyocytes showed that excess palmitate activated the mTOR-C1 complex and subsequently attenuated PGC1α-PPARα transcription of β-oxidation and mitochondrial electron chain factors, affecting mitochondrial/glycolytic ATP synthesis. The results suggest that the coexistence of HTN and T2DM induced more significant alterations in cardiac proteins than HTN alone, with HTN-T2DM subjects exhibiting a dramatic downregulation of mitochondrial respiration and lipid metabolism. The mTOR-C1-PGC1α-PPARα axis may serve as a potential therapeutic target for future therapies. Overall this is an interesting study. But the results are too preliminary, and many open questions and flaws prevent acceptance in its current form.

Please show the full name of CABG, DAPT, for the first time in line 73.

Please correct the typing mistake of Table 3: PPARâ–¡, PGC-1â–¡, and other typing mistakes.

Please replace the clear image of the Figures. The quality of the image could be better and must improve.

Please provide the approved Institutional Review Board (IRB) number by the Ethics Committee of Clinical Investigation of Doce Octubre and Fundación Jiménez Díaz Hospitals.

Author Response

Thanks for reviewing our manuscript. Here we answer point by point all required items. The changes have been included (in red) in the latest version of the work.

Sincerely,

Óscar Lorenzo

Referee #1:

The manuscript by Hang et al. et al. describes with type-2 diabetes (T2DM) and arterial hypertension (HTN) are the significant risk factors in Heart failure. Proteomics and bioinformatics of cardiac biopsies from patients with or without T2DM/HTN revealed substantial changes in 677 proteins. HTN-T2DM had 529, and HTN had 41 proteins altered versus control, with 81% of HTN-T2DM proteins distinct from HTN. Bioinformatics implicated mTOR signaling and reduced activation of AMPK/PPARα, with PGC1α, fatty acid oxidation, and oxidative phosphorylation regulation. In vitro studies using cultured cardiomyocytes showed that excess palmitate activated the mTOR-C1 complex and subsequently attenuated PGC1α-PPARα transcription of β-oxidation and mitochondrial electron chain factors, affecting mitochondrial/glycolytic ATP synthesis. The results suggest that the coexistence of HTN and T2DM induced more significant alterations in cardiac proteins than HTN alone, with HTN-T2DM subjects exhibiting a dramatic downregulation of mitochondrial respiration and lipid metabolism. The mTOR-C1-PGC1α-PPARα axis may serve as a potential therapeutic target for future therapies. Overall this is an interesting study. But the results are too preliminary, and many open questions and flaws prevent acceptance in its current form.

Thank you for your evaluation. We agree that data could be preliminary though very interesting since the proposed pathway can be targeted by new therapies for HTN and T2DM patients with cardiovascular damage. Now, we have added new experiments to justify the mTORC1-PGC1a-PPARa axis as at least partial responsible for the deleterious injury in these hearts. After silencing PGC1a in cultured cardiomyocytes it further reduced the mitochondrial bioenergy under high fatty acid incubation. The lack of PGC1a expression diminished the total ATP and both mitochondrial and glycolytic ATP. Also, a similar effect was found after pharmacological inhibition of PGC1a with SR-18292. In contrast, a PGC1a activator (ZLN005) improved some PGC1a-PPARa-dependent mitochondrial genes such as PPARa and the key component of the complex II, SDHB. These new data have been added in Figure 6. The previous assay of bioluminescence for ATP quantification is now in Suppl. Fig 3A.

Please show the full name of CABG, DAPT, for the first time in line 73.

Thanks for the annotation. The full names of CABG and DAPT were already included in the Methodology, which appeared later in section 4 following the journal template. Now, we have changed the order of sections, and Methodology comes earlier. CABG and DAPT names appear in lines 78 and 88.

Please correct the typing mistake of Table 3: PPARâ–¡, PGC-1â–¡, and other typing mistakes.

Sorry for the mistakes. We have corrected these words and other mistakes along the manuscript. Thanks

Please replace the clear image of the Figures. The quality of the image could be better and must improve.

Again, sorry for the inconvenience. After copy-pasting figures on the template document they significantly lost quality. Now we have included better resolution figures, which have been also independently uploaded on the platform. Thanks

Please provide the approved Institutional Review Board (IRB) number by the Ethics Committee of Clinical Investigation of Doce Octubre and Fundación Jiménez Díaz Hospitals.

Yes, thanks, the reference number has been included.

Reviewer 2 Report

The manuscript entitled “Potential role of the mTOR-C1-PGC1α-PPARα axis under type-II diabetes and hypertension in the human heart” investigated the cellular mechanism of heart dysfunction following HTN or HTN+T2DM. The question is a very important in the field of cardiovascular diseases. They used high throughput analysis to perform a mechanistic study which is very valuable. The results of high throughput study were validated in an in vitro system. The experimental design was very sophisticated. The results presentation and interpretation was very convincing. And the findings were discussed quite sufficient. In general, it is of high interest for cardiologists and cardiovascular researchers and I recommend its publication in the IJMS.

There are only few minor points which I listed below:

1.  The in vitro experiments were not explained adequately in the abstract.

2.  In the abstract line 39, the sentence refers to human samples. However, it should be in in vitro model of HTN-T2DM and its results.”.  

3.  In the methods section, the RT-qPCR looks new and uses fam-fluorophores. Is it a different protocol than using primers.

4.  In the methods section, all antibodies’ information including the manufacturer and batch number should be mentioned.

5.  In the methods section, the version of soft wares that were used, specifically in bioinformatics analysis, should be mentioned.

6.  Some English editing is required. For instance, dyslipidemia, purchased from ThermoFisher and so on

Author Response

Thanks for reviewing our manuscript. Here we answer point by point all required items. The changes have been included (in red) in the latest version of the work.

Sincerely,

Óscar Lorenzo

Referee #2

The manuscript entitled “Potential role of the mTOR-C1-PGC1α-PPARα axis under type-II diabetes and hypertension in the human heart” investigated the cellular mechanism of heart dysfunction following HTN or HTN+T2DM. The question is a very important in the field of cardiovascular diseases. They used high throughput analysis to perform a mechanistic study which is very valuable. The results of high throughput study were validated in an in vitro system. The experimental design was very sophisticated. The results presentation and interpretation was very convincing. And the findings were discussed quite sufficient. In general, it is of high interest for cardiologists and cardiovascular researchers and I recommend its publication in the IJMS.

There are only few minor points which I listed below:

  1. The in vitro experiments were not explained adequately in the abstract.

Thanks, we agree, and we have explained more in detail those experiments (lines 31-33). Cultured rat cardiomyocytes were used for analysis (protein level and activation, mRNA expression, and bioenergetic performance) of key molecular mediators under stimulation with main components of HTN and T2DM (high -glucose and/or -fatty acid, and angiotensin-II.

  1. In the abstract line 39, the sentence refers to human samples. However, it should be in in vitro model of HTN-T2DM and its results.”.

Yes, thanks, both human and in vitro studies indicated that mitochondrial alteration could be key for cardiac dysfunction under HTN and T2DM.

  1. In the methods section, the RT-qPCR looks new and uses fam-fluorophores. Is it a different protocol than using primers.

Thanks for the comment. Not really, the protocol is similar. In the new and more sensitive approaches to evaluate the amount of messenger RNA by quantitative PCR (qPCR), the gene expression assays are used as a primer pair labelled with different fluorophores like FAMTM, with different absorption spectrum.

  1. In the methods section, all antibodies’ information including the manufacturer and batch number should be mentioned.

Yes, we agree, this information could be interesting for readers. We have added that data in lines 163-168. Thanks

  1. In the methods section, the version of soft wares that were used, specifically in bioinformatics analysis, should be mentioned.

Yes, it will be interesting. The software versions have been added along Methodology section, thanks.

  1. Some English editing is required. For instance, dyslipidemia, purchased from ThermoFisher and so on

Thanks, we have revised and corrected the language.

Reviewer 3 Report

The authors of “Potential role of the mTOR-C1-PGC1α-PPARα axis under type-2 II diabetes and hypertension in the human heart” claim to have performed proteomics on human heart biopsies of patients with HTN or T2DM and controls.

This manuscript has been rather painful to read due to major design mistakes, many minor mistakes and inaccuracies as well as lack of details in the appropriate moments. Due to the vast amount of errors I was not able to finish reading the manuscript.

General comments:

Major concerns:

-        Again, this manuscript has been rather painful to read due to many minor mistakes, inaccuracies and lack of details in the appropriate moments, low-resolution figures and many more....

-        This is a study investigating differences between HTN and HTN+T2DM in coronary heart disease with preserved ejection fraction …. This is not even really mentioned.

-        Cardiac biopsies from T2DM only are missing as a control group and would have been nice

-        Male: female ratio is completely off…

-        The control group is bad with a completely different disease (valve disease)

-        One patient with HTN had LV hypertrophy? This must be wrong… This means you are not even looking at real HTN but beginning of well-controlled HTN – this suggests you are not even investigating the claimed disease/risk factor

Other general concerns:

-        They should change their rather unconventional abbreviations like Gluc. Me., FA Me., B-ox, Oxid., Inflam. Especially HF is an unfortunate abbreviation as it is very commonly used for heart failure!

Abstract:

-         “discovery of key molecular signalling may suggest new targets for therapy.” You mean common and/or overlapping key molecular signaling pathways ?

-        Cultured cardiomyocytes of what? Human? Murine? Sentence stands in no relationship with the sentences before and after that… makes no sense…

-        Cardiopathy? You mean cardiomyopathy? Or heart disease? 

-        Suddenly preserved ejection fraction ? were all biopsies from preserved EF? Are you suggesting this is HFpEF? Which disease entity are we examining here? What was the reason for surgery? All not clearly defined in the abstract

-        After filtering out …

-        “Dramatic” is not a very scientific term

Introduction:

-        PMID: 36856044 is a potent study and should be included – seems to directly contradict your hypothesis

Results:

-        This is the first time that the underlying disease is mentioned!! Ischemic cardiomyopathy or coronary artery disease with preserved ejection fraction !!!

-        6 male, 1 female? In this case it is likely better to only examine males….

-       

Cannot continue reading … too much wrong here…

Figures:

In general:

-        At least within the manuscript all figures need a higher resolution – it is a miracle to me that basic IT knowledge cannot be applied to have readable figures

-        Bar graphs are not sufficient nowadays. Please provide (additional) individual dots

-        Tables missing n.s. or significance between groups

Specific:

-        Fig. 1 has to be changed, maybe a clustering dot plot or something; circle-colour code (A) is hard to understand and follow for B

-        Fig. 3: Bars with dots (so you can see the # samples) & error bars are incomplete

-        Fig.3 & 4: which # group is compared to which?

-        Fig.4 & 5 B: #p<0.05 vs. HF. Where is HF in the graph? What does * mean?

-        Fig. 5. Legend, which one? Should be under the figure; Ctrl groups gene expression SEM is missing. Does it mean one sample then?

-        Fig.6B: what does XP ATP rate index mean or xp? Seahorse? Because we would not use XP the in the legend (y axis). Does it mean something different? It just says seahorse in the legend under the figure not XP seahorse. Why is there a change of error bars from Fig.6A to B? It should be the same and look like B (SEM); SEM or SD?

-        Fig. 8: mTOR-C1 and mTOR-C2? No common abbreviations: rather mTORC1 …

-        Suppl. Fig 1 same for Rap (SEM missing)

-        *<0.05? looks highly significant; some lines are smaller and some lines are bigger within one bar?

-        Suppl. Table 1: why table, a figure or graph would be nicer. Also the table is too long. Why did they put it into the supplements; rRNA maturation. Why is there a full stop (page 25;page 28); p 27 space is missing before the round bracket; page 28 there is an additional dash

-        Table 1B: BMI: which one has a non-pathological condition and which one not. It’s not clear. Was does non-pathological condition mean? What does the (y-o) next to age mean? Why would they say 85.6% male? It’s 6 male and 1 female?

-        Table 2: you can read the numbers, why the colour code?  What’s the z-score and where is the explanation for the table. There is no table 2 in the paper. Why?

-        No relation between cell culture experiments and proteomics & bioinformatics

-        What’s the additional value of this studies/experiments for the research? It just says that this pathway suggests new targets for therapy. There are no experiments to support this. There are several studies which show experiments with an additional value related to the mTOR pathway T2DM & HTN

Patents:

-        Under Point 7 (patents) there is everything but patents

Author Response

Thanks for reviewing our manuscript. Here we answer point by point all required items. The changes have been included (in red) in the latest version of the work.

Sincerely,

Óscar Lorenzo

Referee #3

The authors of “Potential role of the mTOR-C1-PGC1α-PPARα axis under type-2 II diabetes and hypertension in the human heart” claim to have performed proteomics on human heart biopsies of patients with HTN or T2DM and controls. This manuscript has been rather painful to read due to major design mistakes, many minor mistakes and inaccuracies as well as lack of details in the appropriate moments. Due to the vast amount of errors I was not able to finish reading the manuscript.

General comments:

Major concerns:

-        Again, this manuscript has been rather painful to read due to many minor mistakes, inaccuracies and lack of details in the appropriate moments, low-resolution figures and many more....

Sorry for the inconvenience. We are unhappy for that, and we hope we can amend it now. Thanks

-        This is a study investigating differences between HTN and HTN+T2DM in coronary heart disease with preserved ejection fraction …. This is not even really mentioned.

Yes, our patients exhibited coronary heart disease without systolic dysfunction. The preservation of systolic function was included in the Abstract, Introduction, Results, Discussion and Conclusions section. Now, we have also added from the beginning the term of coronary heart disease. Thanks for this important clarification.

-        Cardiac biopsies from T2DM only are missing as a control group and would have been nice

Yes, we agree with your comment. In fact, a T2DM group was initially included in the design of the study. Unfortunately, it is very unusual that patients undergoing cardiac surgery have isolated T2DM without hypertension. Thus, we only could recruit two patients with T2DM only, which was not enough to obtain consistent results. We already included that issue in the Limitation of the Study section of the paper. Thanks

-        Male: female ratio is completely off…

The male % was already included in Table 1B.

-        The control group is bad with a completely different disease (valve disease)

As it could be expected, it was very hard to find patients undergoing CABG in the absence of both hypertension and T2DM. As you know, those conditions are key risk factors for the development of CAD. Furthermore, the existence of T2DM favors the choice of CABG as an interventional therapy. Thus, our control group included patients without HTN, T2DM and systolic dysfunction, and mainly undergoing valve replacement.

-        One patient with HTN had LV hypertrophy? This must be wrong… This means you are not even looking at real HTN but beginning of well-controlled HTN – this suggests you are not even investigating the claimed disease/risk factor.

We do not understand very well this comment. Indeed, one patient with hypertension showed LV hypertrophy. As you know, LV hypertrophy is a consequence of hypertension, and this was not an exclusion criterium of the work.

Other general concerns:

-        They should change their rather unconventional abbreviations like Gluc. Me., FA Me., B-ox, Oxid., Inflam. Especially HF is an unfortunate abbreviation as it is very commonly used for heart failure!

Ok, we agree and have changed HF (by HFA) and the rest of abbreviations along manuscript and figures. Thanks

Abstract:

- “discovery of key molecular signalling may suggest new targets for therapy.” You mean common and/or overlapping key molecular signaling pathways?

Yes, you are right. We included the term “common” in the sentence. Thanks

- Cultured cardiomyocytes of what? Human? Murine? Sentence stands in no relationship with the sentences before and after that… makes no sense…

Yes, we agree. We have changed the whole sentence by “Also, cultured rat cardiomyocytes were used for analysis (protein level and activation, mRNA expression, and bioenergetic performance) of key molecular mediators under stimulation with main components of HTN and T2DM (high -glucose and/or -fatty acid, and angiotensin-II)”. Thanks

- Cardiopathy? You mean cardiomyopathy? Or heart disease? 

We have changed to coronary heart disease. Thanks

- Suddenly preserved ejection fraction? were all biopsies from preserved EF? Are you suggesting this is HFpEF? Which disease entity are we examining here? What was the reason for surgery? All not clearly defined in the abstract

Yes, we have included an introductory sentence in the abstract: “Intraoperative cardiac biopsies were obtained from patients with coronary heart disease and preserved systolic function, with or without HTN and/or T2DM, who underwent coronary artery bypass grafting (CABG).” Thank you.

-  After filtering out …

We have changed to “filtering”, thanks for that.

- “Dramatic” is not a very scientific term

Dramatic has been replaced by more appropriated scientific words, thanks.

Introduction:

- PMID: 36856044 is a potent study and should be included – seems to directly contradict your hypothesis

Thanks for this suggestion. We would say that this paper may reinforce our finding. The cardiac ATP may be originated by multiple energy substrates mostly fatty acids and carbohydrates, but in heart failure, with or without reduced ejection fraction, an impaired metabolic flexibility has been described. In fact, our HTN-T2DM hearts showed a reduction in fatty acid oxidation enzymes and in downstream pathways (TCA, mitochondrial electron chain), but also in glycolysis. In HFA-stimulated cardiomyocytes, ß-oxidation and mitochondrial ATP production were also decreased. Interestingly, these cardiomyocytes did not switch to glycolytic metabolism either, suggesting a metabolic inflexibility under HFA (at least at 90 uM for 24h). In this sense, we have added new data in Figure 6 about mitochondrial bioenergy. By both silencing and pharmacological inhibition of PGC1a under HFA, the total ATP production and both mitochondrial and glycolytic ATP were further reduced. In contrast, exogenous activation of PGC1a under HFA was able to recover the PGC1a-dependent gene expression. The Discussion section has been really enriched with this comment, and new references have been added. Thanks for helping us.

 Results:

- This is the first time that the underlying disease is mentioned!! Ischemic cardiomyopathy or coronary artery disease with preserved ejection fraction !!!

Yes, sorry, we have now established from the beginning of manuscript that patients exhibited coronary artery disease with preserved ejection fraction. Thanks

- 6 male, 1 female? In this case it is likely better to only examine males….

We have the same ratio of male:female in both HTN and HTN-T2DM groups and we needed 7 samples of each group to better obtain a significant statistic power by proteomics. Thanks

Cannot continue reading … too much wrong here…

Figures:

In general:

-  At least within the manuscript all figures need a higher resolution – it is a miracle to me that basic IT knowledge cannot be applied to have readable figures

Sorry again. After copy-pasting on template, they lost quite significant quality. The figures have been now placed with higher quality. Thanks

- Bar graphs are not sufficient nowadays. Please provide (additional) individual dots

Ok, done. Thanks

- Tables missing n.s. or significance between groups

Ok, done. Thanks

Specific:

-  Fig. 1 has to be changed, maybe a clustering dot plot or something; circle-colour code (A) is hard to understand and follow for B

Yes, we agree. We have added new terms in the figure to clarify.

- Fig. 3: Bars with dots (so you can see the # samples) & error bars are incomplete

The dots have been included and the error bars have been completed, thanks.

- Fig.3 & 4: which # group is compared to which?

Yes, it was explained in the legends. In figure 3A, **p<0.01 vs. control, #p<0.05 and ##p<0.01 vs. HFA, and in figure 3B #p<0.05 vs. HFA (90 μM). In figure 4A, *p<0.05 vs. control, #p<0.05 and ##p<0.01 vs. HFA, an in figure 4B *p<0.05 vs. control, #p<0.05 vs. HFA.

- Fig.4 & 5 B: #p<0.05 vs. HF. Where is HF in the graph? What does * mean?

In Figure 4A, 4B, and 5A, HFA is just next to the left side of Rapamycin. The * means p<0.05 vs. control.

- Fig. 5. Legend, which one? Should be under the figure; Ctrl groups gene expression SEM is missing. Does it mean one sample then?

Yes, the legend was placed just behind Figure 5. Now you can read it in lines 427-431. No, in the qPCR assays, we show the data in n-fold. Following the delta-delta-Ct method, control values from 5 experiments were fixed at 1, and the stimuli are showed as n-fold over control.

- Fig.6B: what does XP ATP rate index mean or xp? Seahorse? Because we would not use XP the in the legend (y axis). Does it mean something different? It just says seahorse in the legend under the figure not XP seahorse. Why is there a change of error bars from Fig.6A to B? It should be the same and look like B (SEM); SEM or SD?

The XF ATP rate index is the ratio of the mitochondrial ATP production and the glycolytic ATP synthesis. This ratio was calculated by the Seahorse Wave Pro Software v.2.6.

Yes, sorry, we have now added in the legend the full name of the Seahorse equipment. We have now changed figure 6. The previous assay of bioluminescence for ATP quantification is now in Suppl. Fig 3A. We have now further analyzed the influence of silencing for PGC1a by Seahorse. Interestingly, the lack of PGC1a further reduced the mitochondrial and glycolytic ATP production under high fatty acid incubation. Also, a similar effect was found after pharmacological inhibition of PGC1a with SR-18292, and a PGC1a activator (ZLN005) recovered  the PGC1a related gene expression (PPARa and SDHB).

In Figure 6, error bars represent standard deviation. Thanks for the comment.

- Fig. 8: mTOR-C1 and mTOR-C2? No common abbreviations: rather mTORC1 …

Ok, done. Thanks

- Suppl. Fig 1 same for Rap (SEM missing)

In this case, the interquartile range was included thought it was very small for rapamycin co-stimulation.

- *<0.05? looks highly significant; some lines are smaller and some lines are bigger within one bar?

Yes, indeed, the significancy reached < 0.01 in some cases. Those symbols have been changed. Thanks

 - Suppl. Table 1: why table, a figure or graph would be nicer. Also the table is too long. Why did they put it into the supplements; rRNA maturation. Why is there a full stop (page 25;page 28); p 27 space is missing before the round bracket; page 28 there is an additional dash

We agree with your comments. The Suppl. Table 1 is too long, and this is why it was placed as supplementary. The other mistakes have been corrected. Thanks

- Table 1B: BMI: which one has a non-pathological condition and which one not. It’s not clear. Was does non-pathological condition mean? What does the (y-o) next to age mean? Why would they say 85.6% male? It’s 6 male and 1 female?

As you know, obesity can be considered from 30 to 34.9 kg/m2 of BMI. Thus, only HTN-T2DM patients were in the pathological range.

The non-pathological ranges for the lipid profile were obtained from the guidelines of the European Society of Cardiology. These ranges help to stratify patients according with known cardiovascular risk factors such as obesity, lipid, and glucose profiles.

Y-o has been changed to years-old.

Yes, 85.7% corresponded to 6:1 ratio for male:female.

Thank you.

- Table 2: you can read the numbers, why the colour code?  What’s the z-score and where is the explanation for the table. There is no table 2 in the paper. Why?

Sorry, but the color scale was only uploaded in the platform. Now is next to the figura and it refers to z-score value. The candidate molecular pathways are depicted for HTN-T2DM and HTN hearts as bias-corrected activation z-scores. A significant activation or inhibition was considered if z-score ≥ 2 (light green-dark green) or z-score ≤ -2 (orange-red), respectively. The non-significant z-scores are in white colour with a dash. Following Ingenuity pathway analysis (IPA®, 2000-2019 QIAGEN), activation z-score represents the bias in gene regulation that predicts whether the upstream regulator exists in an activated (positive values) or inactivated (negative values) state, based on the knowledge of the relation between the effectors and their target molecules. Only values with |Z-score| ≥ 2 were considered. With respect to the graphical representation of the data, a red/green color code was used for proteins detected in the proteomics experiment, red proteins being downregulated, and green proteins upregulated. IPA correlates the expression pattern of the proteins under analysis with the information contained in its knowledgebase to predict the state of activation/inhibition of these upstream regulators/biological processes. This has been updated also in the Methodology, thanks.

- No relation between cell culture experiments and proteomics & bioinformatics.

By proteomics, we discovered key altered proteins (mostly downregulated) in HTN and HTN-T2DM hearts. Most of them were metabolic, mainly under the T2DM coexistence. Among these factors, most of them belonged to mitochondrial respiration, and interestingly, proteins from the lipid metabolism were only differentially expressed (reduced) when we compared HTN-T2DM with HTN. By bioinformatics, the major predicted downregulated pathway was the oxidative phosphorylation, followed by the TCA, AMPK, and fatty acid oxidation. About upstream regulators, the IPA predicted the highest positive z-score for Rictor (a key mTORC2 component) and high negative z-scores for PGC1a and PPARa. Thus, we checked the mTOR signaling and related factors (AMPK, PGC1a, PPARa, ACADm, SDHB) in cultured cardiomyocytes under excess of lipid (as well as glucose and angiotensin-II). Then, we focused on the potential PGC1a role on mitochondrial bioenergetics and related gene expression.

- What’s the additional value of this studies/experiments for the research? It just says that this pathway suggests new targets for therapy. There are no experiments to support this. There are several studies which show experiments with an additional value related to the mTOR pathway T2DM & HTN

We propose the mTORC1-PGC1α-PPARα pathway to be targeted by new therapies for HTN and T2DM patients with cardiovascular heart disease and preserved systolic function. By high throughput analysis and in vitro validation, we have performed a mechanistic study to demonstrate the implication of the mTORC1-PGC1α-PPARα axis in this pathology. The cellular mechanism of heart dysfunction following HTN or HTN+T2DM is a current question to be solved in the field of cardiovascular diseases, and this signaling could be of high interest for cardiologists and cardiovascular researchers in a near future. In this regard, after silencing PGC1a the mitochondrial bioenergy was further reduced under high fatty acid incubation. The lack of PGC1α expression diminished the total ATP and both mitochondrial and glycolytic ATP. Also, a similar effect was found after pharmacological inhibition of PGC1a with SR-18292. An unbalanced mitochondrial metabolism can trigger pro-oxidative and inflammatory responses, and the lack of ATP could trigger contractile dysfunction. However, a PGC1α activator (ZLN005) improved some PGC1a-PPARa-dependent mitochondrial genes such as PPARα and SDHB. Likely, by recovering PGC1a function we could be able to improve and help to balance mitochondrial metabolism, which might positively influence on cardiac performance in these patients.

Thanks for the comment

Patents:

-  Under Point 7 (patents) there is everything but patents

The template from the journal included this section that have been now deleted. Thanks

Round 2

Reviewer 1 Report

no more request 

Author Response

Thank you

Reviewer 3 Report

Dear authors, the manuscript is now much clearer in its presentation. Nevertheless, I have minor comments to consider.

Table 1B: TG seem significantly different and this should be marked with *

Table 1C: presentation has really no meaning. you should show % for each drug in the respective groups, and again you should mark significant differences. Ideally, as mentioned before you also mark non significant differences with "ns" for example. Otherwise, we have to assume that you have not checked for significance, which I am sure you did

In table 3 you show upregulation of mTORC2 (RICTOR) in human hearts, but in Figure 2 you investigate targets of mTORC1 (pP70). Maybe you can show RAPTOR (mTORC1) increase or PRAS40 decrease in human hearts, if applicable? Or show mTORC2 targets like FoxO1, NDRG2, SGK,... in cardiomyocytes in vitro? At least you could mention the known link of mTORC1 and 2 (PMID: 24008870) and the known mTORC1 effect on hypertrophy and metabolic substrate utilization (PMID: 33401933 and 32393148 and others), and its known potential pharmaceutical benefit (PMID: 32393148).

Thanks!

    Author Response

    Thanks for reviewing again our manuscript. Here we answer point by point all required items. The changes have been included (in red) in the latest version of the work.

    Dear authors, the manuscript is now much clearer in its presentation. Nevertheless, I have minor comments to consider.

    Table 1B: TG seem significantly different and this should be marked with *

    Yes, the TG levels in HTN-T2DM were significantly higher than control, but not than HTN. The * has been included. Thanks for this detail. 

    Table 1C: presentation has really no meaning. you should show % for each drug in the respective groups, and again you should mark significant differences. Ideally, as mentioned before you also mark non significant differences with "ns" for example. Otherwise, we have to assume that you have not checked for significance, which I am sure you did

    Ok, yes, we agree. We have included more information by adding the % and statistical significance. The legend and statistical methodology have been also modified. Thank you.

    In table 3 you show upregulation of mTORC2 (RICTOR) in human hearts, but in Figure 2 you investigate targets of mTORC1 (pP70). Maybe you can show RAPTOR (mTORC1) increase or PRAS40 decrease in human hearts, if applicable? Or show mTORC2 targets like FoxO1, NDRG2, SGK,... in cardiomyocytes in vitro? At least you could mention the known link of mTORC1 and 2 (PMID: 24008870) and the known mTORC1 effect on hypertrophy and metabolic substrate utilization (PMID: 33401933 and 32393148 and others), and its known potential pharmaceutical benefit (PMID: 32393148).

    Thanks!

    Yes, we believe that under HTN and T2DM, some stimuli (i.e., excessive fatty acid) can trigger a cardiac imbalance in the mTOR signaling, leading to higher stimulation of the mTORC1 branch and associated deleterious effects. In this sense, mTORC1 activation has been related either to cardiac hypertrophy and depressed autophagy and to deviation of fatty acid oxidation to glycolysis to confer protection against ischemia-reperfusion injury. As potential, perhaps temporary response, the heart could activate adaptive mechanisms which include upregulation of mTORC2 complexes and subsequent mediators (i.e., FoxO1). The link and counter-regulation of mTOR complexes have been previously suggested. For instance, predominant mTORC1 signaling mediated by suppression of mTORC2 with Rictor increased cardiomyocyte apoptosis and tissue damage after myocardial infarction. However, shifting toward mTORC2 signaling by inhibition of mTORC1 with a mTORC1-regulating subunit, PRAS40, reversed these effects. Intriguingly, specific mTORC1 inhibition by PRAS40, but not by rapamycin, improved cardiac metabolic function, blunted hypertrophic growth, and preserved systolic function in experimental diabetes. These comments and related references have been added to the Discussion section. Thanks for this interesting comment. Further investigations on mTOR subunits with regulation capacities, such as PRAS40 or Raptor for mTORC1, and Rictor or Deptor, for mTORC2 are very interesting for our future studies.